# Gut Microbiome Composition in Patients with Chronic Urticaria: A Review of Current Evidence and Data

**DOI:** 10.3390/life13010152

**Published:** 2023-01-04

**Authors:** Mirela Krišto, Liborija Lugović-Mihić, Melba Muñoz, Maja Rupnik, Aleksander Mahnic, Petar Ozretić, Morana Jaganjac, Diana Ćesić, Matea Kuna

**Affiliations:** 1Department of Dermatovenereology, University Hospital Centre Sestre Milosrdnice, 10000 Zagreb, Croatia; 2School of Dental Medicine, 10000 Zagreb, Croatia; 3Institute of Allergology, Charité-Universitätsmedizin Berlin, Freie Universität Berlin, and Humboldt-Universität zu Berlin, 13125 Berlin, Germany; 4Fraunhofer Institute for Translational Medicine and Pharmacology ITMP, Allergology and Immunology, 13125 Berlin, Germany; 5Department for Microbiological Research, National Laboratory for Health, Environment and Food, SI-2000 Maribor, Slovenia; 6Department of Microbiology, Faculty of Medicine, University of Maribor, SI-2000 Maribor, Slovenia; 7Laboratory for Hereditary Cancer, Division of Molecular Medicine, Ruđer Bošković Institute,10000 Zagreb, Croatia; 8Laboratory for Oxidative Stress, Ruđer Bošković Institute, 10000 Zagreb, Croatia

**Keywords:** gut microbiome, intestinal microbiome composition, chronic spontaneous urticaria, dysbiosis, metabolome

## Abstract

Recent studies have linked gut microorganism composition and chronic urticaria (CU); however, the underlying mechanisms responsible for this connection are unknown. Since the human immune system is in homeostasis with microbiota, and the composition of the microbiome regulates the development and function of the immune system, it is likely that an alteration of microbiota components (a dysbiosis) could influence the course of chronic spontaneous urticaria (CSU), including disease severity, patient quality of life and treatment outcome. To date, several studies have identified changes in the gut microbiota composition of patients with CSU, though only a few have exhibited metabolic abnormalities associated with gut dysbiosis. The studies on CSU patients predominantly showed that the relative abundance of beneficial bacteria was decreased (*Firmicutes* and *Bacteroides*), while that of opportunistic bacteria was increased (*Enterobacteria* and *Proteobacteria).* In addition, serum metabolome analysis revealed that gut microbiota-associated alterations in unsaturated fatty acids and the butanoate metabolism pathway may play a role in CSU. These findings are potentially associated with inflammation mediated by the imbalance of Th1/Th2/Th17 cytokines, which might contribute to CSU pathogenesis. Further research in this field could improve clinical, diagnostic, and therapeutic approaches to patients with CSU. By applying new knowledge on gut microbial communities and metabolomics, future CSU therapies could modify the microbiota composition using agents such as probiotics or other similar agents, which, in combination with current standard therapies, could hopefully lead to a reduction in symptoms and an improved quality of life for CSU patients.

## 1. Introduction

Chronic urticaria (CU), whether spontaneous or inducible, is characterized by recurrent episodes of pruritic wheals, with or without associated angioedema persisting longer than 6 weeks [1,2]. Globally, up to 1% of the general population suffers from CU at some point in life, and a trend of increasing prevalence has been observed in recent years [3]. Chronic spontaneous urticaria (CSU), especially concomitant angioedema, greatly impacts patients’ work, studies, quality of life and mental health, and it also imposes a huge economic and social burden on families and society [1,2,3,4].

Although the course of CSU can be self-limiting, in 10–25% of patients it lasts longer than 5 years [5]. Treatment is often focused on “symptom control”. Patients with a prolonged disease duration, or therapy-refractory symptoms, experience additional quality of life burdens. The current treatment of CU is largely dependent on symptomatic treatment with second-generation non-sedating H1-antihistamines (nsAHs) [1,2]; however, 40–55% of patients are unresponsive to conventional doses of antihistamines [6,7]. Clinically, resistant CSU (rCSU) is defined by unresponsiveness to nsAHs after 2 to 4 weeks of treatment at approved dosages [8]. According to current guidelines, therapeutic options for rCSU include increasing doses (up to four-fold of the approved dose) of nsAHs, and add-on medications: glucocorticoids, cyclosporine and biological agents (omalizumab) [9], often with limitations due to the possibility of serious side effects, unresponsiveness, and the high cost to the healthcare system [1,2]. Thus, there is an urgent need to find novel diagnostic tools and alternative therapies [1,2,10]. Unravelling the pathogenesis of CSU will represent a step forward in the management of CSU patients.

Recent studies have reported an altered gut microbiota composition in patients with CSU [11,12,13]. It has been hypothesized that pro-inflammatory responses caused by alterations in the gut microbiome, mediated by the imbalance of Th1/Th2/Th17 cytokines, might contribute to the pathogenesis of CSU [14]. Additionally, recent evidence from human metabolomics has shown enteric dysbacteriosis in CSU subjects compared to healthy subjects. Thus, uncovering potential causes of inflammation might provide new strategies for improving symptom control and minimizing the disease’s burden on CSU patients [15,16,17].

The purpose of this review is to present current knowledge on the characteristics of the gut microbiome and its possible association with CU, taking into consideration that therapeutic effects on microbiome dysbiosis could improve CU symptom control. We use data from studies found on Pubmed, as well as other meaningful literature. In our search, we used the key words “chronic urticaria” and “gut microbiota”; we analysed and included articles published before August 2022.

## 2. Immunopathogenesis of Chronic Urticarial and Immune Changes in the Peripheral Blood and Skin of Patients

The pathogenesis of CU is poorly understood, especially CSU, but current evidence suggests that most CSU cases have an autoimmune etiology [18]. Understanding these principles is complex, and according to current knowledge, aside from humoral (autoantibodies), it also involves cellular responses (auto-reactive T cells) and the dysfunction of adaptive cellular immunity [19,20]. According to current literature data, crucial pathogenetic factors/events for CSU are the activation of mast cells, basophils, T cells, and eosinophils, as well as increased vascular permeability and vasodilation, activation of coagulation pathways and autoimmunity processes, and finally, the release of mast cell mediators [21]. Thus, CSU involves the participation of both innate immunity (mast cells, basophils, neutrophils, eosinophils, monocytes, macrophages, natural killer [NK] cells, innate lymphocytes [ILC], complement factors) and adaptive immunity (Th1 and, Th2 cells, Th9 cells, Th17 cells, regulatory T cells [Treg cells], B cells, antibodies), wherein mast cells link innate and adaptive immunity. Mast cells have various roles: they induce/modulate T-cell activation and polarization; stimulate B cells to produce IgE (through IL-4 and IL-13); produce mediators (e.g., TNFα), which act on vascular endothelial cells’ up-regulation and adhesion molecules‘ expressions, which promote the recruitment of T cells; allergen-specific Th2 cells stimulate B cell production of IgE, which stimulates/activates mast cells and basophils [21]. In the skin tissue of urticarial CSU lesions, mast cells predominate, and there are increased numbers of basophils, eosinophils, macrophages, neutrophils and T cells, while there are decreased numbers of Th17 cells. In the serum of CSU patients, decreased numbers of Treg cells (among PBMC) are found, while increased IL-17 serum levels are seen (produced by Th17) [21]. Mast cells and T cells interact through various contacts, including through their inflammatory mediators and their receptors. It is also important that T cells express histamine receptor H4R (known for its role in inflammation and allergies), which activates Th2 and Th17 cells. Thus, histamine mediates the enhancement of Th2 cytokine secretion while inhibiting Th1 cytokine production, maintaining Th1/Th2 cell balance and supporting the dominance of Th2 cells. In addition, mast cells produce cytokines that promote Th2 cell activation, survival and migration (PGD2 and leukotriene E4) [21].

Research in this field has illuminated the previously unrecognized role of pro-inflammatory Th17 cells and the immune imbalance between them and Treg cells. It is therefore important here to mention genetic factors and expressions of genes. In a study on CSU patients by Prosty et al., several upregulated Th17-related pathways were identified, including IL-17 signaling, Th17 cell differentiation, IL-6 and IL-23 mediated signaling events in lesional samples versus samples from both non-lesional and healthy persons [22]. Key Th17-related genes (e.g., IL-6) were also upregulated in lesional CSU samples compared to samples from non-lesional and healthy persons [22]. Aside from this, a study by Atwa et al. has shown high serum levels of IL-17, IL-23 and TNF-α in CSU patients compared to controls, positively correlating with disease activity (UAS7 scores) [23]. Additionally, a recent open label trial documented that secukinumab, an IL-17α inhibitor, significantly improved symptoms of refractory CSU, which underscores the therapeutic relevance of the Th17 pathway in CSU [24]. Concerning the skin, another interesting study by Sabag et al. revealed increased cutaneous CD4+ T cells and mast cells infiltration in CSU patients (in both lesional and non-lesional skin) compared to healthy controls [24]. These findings in the skin of CSU patients versus healthy controls were supported by a study from Toubi et al. that showed autoimmunity in CSU involving both humoral and cellular responses [25]. Notably, mast cell degranulation is mainly thought to be the result of activated mast cells coming into proximity with autoreactive T cells. When IgE antibodies to thyroid antigens or anti-FcɛR1 on mast cells are seen, it might be possible that the autoreactive T cells could have responded to those antigens.

On the other hand, several reports have documented decreased Treg cells in CSU blood versus the blood of healthy controls, which may imply these cells play a role in the CSU autoimmunity [26,27]. Additionally, Treg cells participate in suppression of mast cell degranulation via the signaling of OX40-OX40L. The absence of suppression of mast cell degranulation may contribute to allergic responses and their severity [28]. The study by Prosty et al. also observed increase of Treg cells and resting mast cells in non-lesional versus lesional skin, which suggests that Tregs might inhibit wheal formation by suppressing mast cells into a resting state [22]. However, Treg suppression can be reversed in the presence of activated mast cells, an abundance of IL-6 and scarcity of Th1/Th2 cytokines, inducing a Th17 response [29].

## 3. Characteristics of the Gut Microbiome of Healthy Individuals

The human microbiota encompasses numerous bacteria and other microorganisms (archaea, fungi, viruses and protozoa) residing in/on the body, including the genomic content of organisms inhabiting a particular site in the human body [30,31]. An adult human’s gut microbiota is mainly inhabited by bacteria from three major phyla: *Firmicutes, Bacteroidetes* and *Actinobacteria*, which together make up more than 90% of total gut bacteria. The human gut microbiome consists of a core microbiome (common to all or the great majority of individuals), and a variable microbiome (unique to sub-groups of individuals depending on lifestyle and physiological differences). An imbalance in the composition, or loss of function, of human microbiota is called dysbiosis [32].

Various factors can be involved in bacterial imbalances, e.g., psychosocial stress, health behaviours, nutrition, social relationships, environmental factors, tobacco and alcohol intake, and prescription drug use (e.g., antibiotics or probiotics) [32,33,34]. An alteration of microbiota components is linked to the pathogenesis of psychiatric diseases, inflammatory bowel diseases, cardiovascular and neurological conditions, allergic diseases, cancers, diabetes mellitus, rheumatoid arthritis, metabolic syndrome, liver diseases, obesity, and kwashiorkor [12,32,35].

The human microbiota has a significant impact on the immune system and is necessary for the evolution and regulation of certain elements of the immune system [36]. Bacterial colonization is essential for the establishment of immunity. Specifically, the homeostasis of microbes and the immune system protects and benefits both the microbes and the host [33]. Dysbiosis, on the other hand can contribute to the development and persistence of different diseases. Any quantitative or qualitative changes to the intestinal microbiome may trigger an inflammatory response followed by tissue damage [33,36].

According to the literature data, when gut bacteria interact with dendritic cells and intestinal epithelium, the signalling pathways of immune effector cells, including macrophages, B cells, NK cells, and T cells are activated [37]. Gut lamina propria at a steady state contains large numbers of two populations of CD4+ T cells, helper Th17 cells and regulatory T (Treg) cells [38]. Therefore, in response to specific components of the commensal microbiota, Th17 cells are induced in the intestinal lamina propria. Treg cells, also important, maintain immune homeostasis and promote immune tolerance to allergens [38]. According to several studies, CU patients have a reduced number and function of Treg cells, and some researchers have speculated that the reduction of some bacteria may affect Treg cells’ role in immunity by inducing Toll-like receptors, leading to CSU (occurrence and persistence) [12,13].

## 4. Approaches to Gut Microbiota Analysis

In recent years, advanced technologies for microbiome analysis have contributed to the understanding of the microbiome’s involvement in the pathogenesis of several autoimmune diseases, including systemic lupus erythematosus, rheumatoid arthritis, Sjögren’s syndrome, Crohn’s disease, Behcet’s disease, type 1 diabetes, psoriasis, and psoriatic arthritis [39,40]. We have also learned more about its connection to allergic disorders such as food allergies [41], atopic dermatitis [42] and asthma [43]. Research on microbiome alterations in autoimmune and inflammatory diseases, CSU among others, is growing. Developments in molecular biology and bioinformatics, as well as the reduction of sequencing costs, have made a significant contribution to microbiome research and enables detection of microorganisms as well as functional genes [44,45]. There are two main approaches to microbiota sequencing: targeted 16S rRNA sequencing, and non-targeted sequencing of the entire genetic material (shotgun method). Thus, it should be noted that the field of metagenomics, the study of the structure and function of entire nucleotide sequences, also plays a role in understanding the gut microbiome.

Additionally, in determining the potential impact of the microbiome at the onset of a disease, it is important to understand the healthy gut microbiome [46,47]. Microbiome research has recently become a wide area of interest in medicine, and it has contributed to greater clinical diagnostic and therapeutic solutions since a number of diseases have been linked to microbiome dysbiosis, i.e., changes in the stability and the content of the intestinal, skin, and oral microbiomes [33]. Modern treatment strategies, such as probiotics, prebiotics, and personalization of diet, have exhibited the successful application of knowledge on the microbiome for therapeutic purposes [48,49,50].

Aside from qPCR, most of the studies included in this paper performed 16S rRNA gene sequencing of fecal samples, and a few used untargeted metabolomics from blood samples to examine the differences in the gut microbiome and metabolites between CSU patients and healthy persons. Studies that used 16S rRNA gene sequencing started in 2019 (16S rRNA sequencing is targeted while the shotgun method is non-targeted sequencing of the entire genetic material). Research findings were not consistent with each other and were limited due to their small sample sizes [11].

A recent combinational analysis looking at the gut microbiome and serum metabolome has shown that changes in unsaturated fatty acids and the butanoate metabolism pathway, related to gut microbiota, might play a role in the pathogenesis of CSU [13,17]. Currently, metabolomics is a compelling area of research that studies the small molecule metabolites that make up an individual’s metabolome. Knowledge about metabolites’ biochemical functions could help us better understand host-gut microbiota interactions. Human phenotype is modulated by the gut microbiome, which was recently found to explain up to 58% of individual plasma metabolite variance [51]. Thus, the gut microbiome is composed of diverse microorganisms that have a protective and structural role in the gut and that also carry out some essential metabolic functions, including the synthesis of amino acids, biotransformation of bile acids and the production of short-chain fatty acids [52]. Changes in the host’s diet, an unhealthy lifestyle or intestinal epithelial cells can induce proinflammatory cytokines; mediators of inflammation, then, dramatically impact the luminal microenvironment, which may consequently result in dysbiosis of the gut microbiome various resulting pathologies, including CSU. Therefore, it is important to understand the interactions between the gut microbiome and the host in both the presence and absence of disease in order to identify undesirable changes in the microbiome associated with the development and progression of CSU. With the development of new technologies, mass spectrometry and nuclear magnetic resonance spectroscopy have emerged as invaluable tools for gut metabolome analysis of various biological matrices, with feces being the most commonly used one. The metabolomics approaches to study gut metabolome profiling have been reviewed in several excellent papers [53,54,55].

## 5. Gut Microbiome in Patients with Chronic Spontaneous Urticaria

The gut microbiota of CU patients has been studied in the past years, but to date there have only been a few studies that have revealed significant differences in gut microbiome composition between CSU subjects and healthy controls (Table 1). Some of these studies show similar results while others exhibit inconsistencies that may be attributed to subjects having different subtypes of CU and to the different research methods used [44,56]. As mentioned above, most studies used 16S rRNA sequencing to identify the species and genera found in healthy individuals and CSU patients and then compare the two, while some only used qPCR to detect the differences in specific species between the two groups [44]. Based on the results, authors from different studies have hypothesized that immune changes in CU are associated with gut microbiome dysbiosis. It is possible that inflammatory responses caused by alterations in the gut microbiome, mediated by the imbalance of Th1/Th2/Th17 cytokines, might contribute to CSU pathogenesis [14].

These findings of gut microbiome alterations in CSU are similar to findings in other allergic and inflammatory diseases such as asthma and atopic dermatitis [35].

For insight into the specific characteristics of the gut microbiome and its microorganisms and bacterial composition, data on alpha and beta diversity for specific diseases, including CSU, is very important. Species diversity in a single sample is alpha diversity. The chao1 value, ACE value, and Shannon index, among others, are methods to measure/express alpha diversity [60]. Beta diversity on the other hand is the ratio of all operational taxonomic units (OTUs); differentiation among groups is identified by analyzing common OTUs [44]. In a study of Zhang et al. from 2021, beta analyses found significant clustering (*p* < 0.001) of controls and CSU patients, meaning the composition of the gut microbiota differed between the two groups [44]. Significant differences (*p* < 0.05) were found in *Proteobacteria* (phylum level), *Bacilli* (class level), *Enterobacterales* (order level), *Enterobacteriaceae* (family level), and *Megamonas, Dialister* and *Megasphaera* (genus level). The authors found that the relative abundance of *Proteobacteria* in CSU patients was significantly greater than in healthy controls. Although *Proteobacteria* are commonly found in the human gut, changes in their relative abundance could cause microbiota dysbiosis and be related to certain conditions and diseases [44]. Other reports showed increased *Proteobacteria* in patients with asthma and allergic diseases [61,62,63].

Lu et al. and Wang et al. reported decreased alpha diversity of gut microbiota in CSU patients, leading to changes in composition of the gut microbiota [11,13]. A study by Wang et al. found an increase in unidentified *Enterobacteriaceae* and a significantly reduced number of *Firmicutes, Bacteroides, Faecalibacterium, Bifidobacterium*, and unidentified *Ruminococcaceae* in CSU patients compared to healthy controls [13]. Lu et al., however, reported a significant decrease in *Bacteroidetes* and an increase in *Actinobacteria* and *Proteobacteria* in CU patients [11]. The results of Liu et al., who studied gut microbiome biomarkers in CSU and symptomatic dermographism (CSD), are similar to those of Wang et al. from 2020 concerning the reduced number of *Firmicutes* in patients with CSU [56]. In contrast, Wang et al. and Zhang et al. reported an increase of *Firmicutes* and *Bacteroides* at the phylum level in CSU patients [17,44]. It is also important to mention that, of beneficial bacteria, *Firmicutes* and *Bacteroidetes*, make up more than 90% in humans [32,64,65,66]. *Firmicutes* breaks down insoluble dietary fibers releasing nutrients and promoting the growth of other species of bacteria in the gut [67]. *Bacteroides* contribute to immune system regulation interact with other microorganisms to promote diverse intestinal flora [68]. Rezazadeh et al. analyzed the stool samples of CU patients and found a relative abundance of *Lactobacillus* and decrease in *Bifidobacterium* [57]. Lu et al. reported that, in patients with CU, the abundance of *Faecalibacterium prausnitzii, Prevotella copri*, *and Bacteroides sp.* was lower, while the abundance of *Escherichia coli* was greater [11].

Additionally, a study by Liu et al. found a lower relative abundance of bacteria from the *Ruminococceae family*, including *Butyricoccus* and *Faecalibacterium*, while subclasses of the *Enterobacteriaceae* family showed a greater relative abundance [56]. In addition, the authors found that bacteria that produce short chain fatty acids in the core microbiota were significantly reduced in CSD patients, including *Subdoligranulum*, which is an obligate anaerobe that can produce butyrate. Short-chain fatty acid production has been shown to enhance Treg cell function [69]. Treg cells are known to suppress inflammatory responses mediated by different T-cell subsets such as Th2 cells. Therefore, Liu et al. propose that impaired Treg cell function can increase Th2 cell numbers and their IL-4 production [56]. Th2 cell-producing IL-4 then leads to IgE production and mast cell activation. It is important to mention that Th2 cytokines IL-4, IL-5, and IL-13 initiate IgE production and therefore, generate IgE antibodies to autoantigens. This is one proposed mechanism of CSU. In addition, IL-4 promotes isotype switching to IgE and this might contribute to generation of IgE anti-IL-24 and IgE anti-TPO, which have been shown to activate mast cells and/or basophils in vitro [70]. Thus, a reduction in short chain fatty acid producing bacteria such as *Subdoligranulum* and *Ruminococcus bromii* can promote mast cell activation and degranulation, leading to the development of hives and itch in CSU patients [56]. In contrast, pathogenic bacteria such as *Enterobacteriaceae* produce lipopolysaccharides known to promote Th2 cell differentiation and IL-4 production [71]. Therefore, it is plausible that increases in *Enterobacteriaceae* trigger urticaria symptoms by activating mast cells and inducing histamine release. Similarly, recent studies on *Ruminococcus bromii*, which is considered a key species for the degradation of resistant starch in the colon, have revealed this bacteria’s protective effect in the case of food allergies [72]. This short chain fatty acid producing bacteria can also produce acetate, which leads to the downstream production of butyrate [56]. Previous studies have shown that butyrate also inhibits inflammation and allergic reactions by inducing the differentiation of Treg cells in vivo and in vitro [69]. Thus, both *Subdoligranulum* and *Ruminococcus bromii* promote the differentiation of Treg cells through different pathways, and their reduced numbers and impaired function might be involved in the pathogenesis of CSD [56].

Of note, bacteria from the *Lachonspiraceae* family, e.g., *Anaerostipes, Blautia* and *Lachnospira*, which were found to be reduced in CSU patients in several recent studies [58,59,73], also produce anti-inflammatory short-chain fatty acids (butyrate and acetate) [74]. Wang et al. show that in CSU patients, α-mangostin and glycyrrhizic acid is increased, and 3-indolepropionic acid, xanthine, and isobutyric acid are reduced [17]. Furthermore, a correlation between intestinal microbiota and metabolites was found, including a positive correlation between *Lachnobacterium* and α-mangostin. A reduced number of bacteria from the *Lachonspiraceae* family in CSU patients could also explain possible Treg and Th2 cell imbalances that lead to higher inflammatory responses. In addition, some studies have assessed the composition of gut microbiota in CSU patients based on their response to nsAHs. Liu et al. found lower abundance of *Lachnospira* in antihistamine responders compared to non-responders. Thus, the authors propose that *Lachnospira* could be a marker for predicting the efficacy of antihistamine use in patients with CSU [59]. Another study by Song et al. also revealed changes in gut microbiota in nsAH rCSU patients; namely, lower bacterial alpha-diversity and similar fecal bacterial communities were found in nsAH rCSU patients compared to those without, which indicates a relation/an association between enteric dysbacteriosis and CSU severity [58]. They found that the greater the abundance of Escherichia and *Prevotella,* the less likely a patient was to experience systemic (not intestinal) inflammation [58].

Recent study results showing that compound probiotic intervention reduced inflammatory responses mediated by the *Escherichia* challenge may suggest that CSU patients refractory to nsAH might benefit from probiotic supplementation [49,73,74,75], though further clinical studies are required to confirm this hypothesis [58]. Thus, in patients with CU, whereas the relative abundance of beneficial bacteria is lower, the abundance of opportunistic bacteria is higher [49,73]. Several studies have found a higher abundance of *Enterobacteria* in CSU patients [11,12,13,44,56]. Anti-inflammatory bacteria such as *Akkermansia muciniphila, Faecalibacterium prausnitzii, Clostridium leptum, genus Lactobacillus* and *Bifidobacterium* have shown to protect from allergic and inflammatory diseases [35]. Anti-inflammatory bacteria might protect against CSU through the induction of Treg cells [76]. Lastly, some studies have shown the beneficial impact of probiotics in CSU patients. *Bifidobacterium* strains are used as probiotics to improve dysbiosis and therefore reduce the inflammatory response [77]. One recent study examined the use of a type of synbiotic (prebiotic + probiotic) named LactoCare in the treatment of CSU and was found to be effective, safe and a satisfactory adjuvant therapy [78].

To confirm current research data, further studies are needed that evaluate microbiome composition and changes in bacteria-derived metabolic products for large cohorts of CSU patients. It remains to be determined whether the microbiome changes observed in CSU patients contribute to the development of CSU or result from changes of the immune response during the course of the disease. Understanding the composition and role of the microbiome in health and disease will shed light on the pathogenesis of inflammatory diseases and might allow the development of novel strategies and therapies that lead to prevention or disease control.

Considering all the above-mentioned pathological mechanisms of the intestine, blood and skin involved in inflammatory and autoimmune processes, it is possible that gut dysbiosis is the initial event in, at least, CSU cases which are considered autoimmune (Figure 1). As mentioned before, depletion of certain bacteria may modulate the immune function of Treg cells by inducing Toll-like receptors of antigen presenting cells on the intestinal epithelium [79]. Moreover, metabolic abnormalities in association with gut dysbiosis, especially changes in unsaturated fatty acids and the butanoate metabolism pathway, contribute to a Th17/Treg imbalance through the inactivation of Treg cells and the upregulation of Th17-related genes. Therefore, correcting Th17/Treg imbalances may be a novel therapeutic approach to CSU by modifying the gut microbiota.

## 6. Therapeutic Implications

Since first-line CSU therapy (antihistamines) may not achieve satisfactory disease control for these patients, there is an unmet need for more effective therapy options, including possible dietary strategies and supplementation [80]. Probiotics are viable microorganisms with beneficial effects on the body and are generally safe when consumed in appropriate quantities [81]. They modulate intestinal microbiota, which also involves the modulation of its immune system, thus improving bowel movement and potentially modulating the IgE response to allergens [82]. Emerging data show benefits of microbiota regulation with probiotics, prebiotics, and symbiotics in various fields of medicine, including dermatology [83]. When an alteration in microbiota components exists (e.g., due to modern lifestyle), probiotic supplementation may counterbalance the Th2 cells activity by promoting Th1 cytokines production and downregulating IgE production via inhibition of IL-4 and IL-5 production [84]. It has been shown that *Lacticaseibacillus paracasei* NC 2461-induced development of CD4+ T cells producing TGF-β and IL-10, which could downregulate IgE production [85]. 

So far, only a few studies on probiotic supplementation for CU patients have been conducted, mostly involving a small number of patients with rCSU. Thus, in a study by Nettis et al., a combination of two probiotics, *Lactobacillus salivarius* LS01 and *Bifidobacterium breve* BR03, was given to CSU patients for 8 weeks (twice daily), which in some patients—most of these patients also suffered from concomitant allergic rhinitis—lead to reduced CSU symptoms and improved quality of life [86]. This study was based on previous research that found that the combination of the two above-mentioned probiotics may reduce the release of pro-Th-2 cytokines from THP-1 cells, favouring an improvement in Th1/Th2 [87]. Therefore, it should be mentioned that some *Bifidobacterium* species have useful anti-inflammatory effects; some strains (used as probiotics) improve dysbiosis and alleviate the inflammatory response, possibly by/through inducing Treg cells with their anti-inflammatory mediators [88].

Another recent randomized controlled trial by Atefi et al. evaluated the efficacy and safety of a synbiotic (prebiotic + probiotic) named LactoCare, which contains high amounts of many acidophilus, *Bifidobacterium breve, Lactobacillus bulgaricus, Bifidobacterium longum, Streptococcus thermophilus*) plus *fructooligosaccharides* as a prebiotic, in CSU treatment [78]. Although the combination of probiotics and antihistamines did not show significant differences in efficacy in this study (in comparison to antihistamines alone), in some patients this combined therapy achieved lower levels of CSU severity, itch and numbers of hives seen. Compared to a previous study by Nettis et al., this recent study by Atefi had clearer enrollment criteria and better positive therapeutic results [78,86]. However, there is still a need for studies with more subjects and improved research designs to obtain significant results and avoid misleading conclusions [78,89]. Generally, probiotics have been shown to be a safe and effective treatment option for resistant CU, mostly reducing urticaria symptoms, such as pruritus, and improving the quality of life of patients with this burdensome disease. Future studies should include a larger number of patients and have a double-blind placebo-controlled design.

## 7. Conclusions

Most studies that have looked at the gut microbiome of CSU patients have observed a loss of microbial diversity in its composition. Microbiome changes lead to dysbiosis and impair immune system regulation. Bacterial products seem to induce an imbalance in proinflammatory and anti-inflammatory T-cell subsets in the gut leading to a shift towards a more proinflammatory phenotype. A proinflammatory response might be responsible for mast cell activation and, therefore, a higher symptom burden in CSU patients. Dissecting the composition of the microbiome in CSU patients is important for the discovery of meaningful biomarkers and therapeutic targets for this burdensome disease.

Diet interventions and functional foods (e.g., probiotics and prebiotics) have already demonstrated benefits for patients with other autoimmune and inflammatory diseases, and the literature discussed here suggest that similar efforts could also correct microbiome dysbiosis in CSU patients. Further studies are needed to support this observation.

## Figures and Tables

**Figure 1 life-13-00152-f001:**
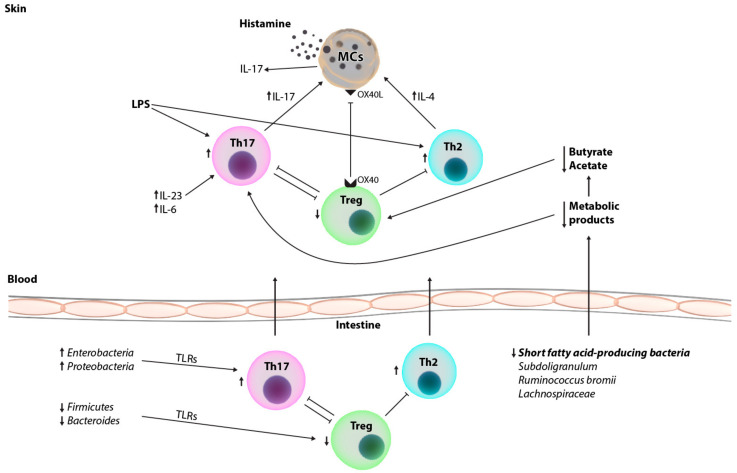
Factors associated with gut microbiome composition characteristics of patients with chronic spontaneous urticaria.

**Table 1 life-13-00152-t001:** A summary of the evidence on the association between gut microbiome composition and chronic urticaria.

Examinees	Methods	Key Results	CU Patients Versus Healthy Controls (HCs)		Limitations	References
			Increase	Decrease		
20 CU patients compared to 20 age- and sex-matched HCs	Bacterial genomic DNA extracted from stool samples and sequenced by PCR; bacterial amounts determined by qPCR.	The frequency and relative amounts of *A. muciniphila*, *C. leptum,* and *F. prausnitzii* were significantly higher in HCs than in those with CU.	Family *Enterobacteriaceae*	Species *Akkermansia muciniphila*, *Clostridium leptum*, *Faecalibacterium prausnitzii* (*p* < 0.001, *p* < 0.01, and *p* < 0.05)	Small number of examinees, more sophisticated analysing methods not included	[12] Nabizadeh et al., 2017.
20 CU patients compared to 20 age- and sex-matched HCs	Fecal samples analysed by PCR for frequency and bacterial load of Lactobacillus and Bifidobacterium genera.	The relative amounts of *Lactobacillus* and *Bifidobacterium* were significantly higher in fecal samples from HCs compared to CU patients.		Genus *Lactobacillus, Bifidobacterium* (*p* = 0.038 and 0.039)	Small number of examinees, more sophisticated analysing methods not included	[57] Rezazadeh et al. 2018.
10 CU patients compared to 10 HCs	Intestinal microbiome was analysed using 16S rRNA sequencing.	Significantly different microbial composition was observed between CU patients and HCs at the genus level.	Phylum *Proteobacteria, Actinobacteria* Order *Enterobacteriales, Lactobacillales, Pseudomonadales;* Genus *Veillonella*, *Sutterella, Streptococcus, Clostridium, and Escherichia;* Species *E.coli*	Phylum *Bacteroidetes* Genus *Faecalibacterium, Prevotella, Lachnobacterium;* Species *Faecalibacterium prausnitzii, Prevotella copri, Bacteroides fragilis, Bacteroides plebeius*	Very small number of examinees	[11] Lu et al., 2019.
100 CSU patients compared to 100 HCs	Fecal and blood samples were analysed using 16S rRNA gene sequencing and untargeted metabolomics, respectively.	The CSU group exhibited differences at the phyla and genera levels with decreased alpha diversity. The serum metabolome analysis revealed changes in unsaturated fatty acids and the butanoate metabolism pathway.	Family *Enterobacteriaceae*	Phylum *Firmicutes;* *Genus Bacteroides, Faecalibacterium, Bifidobacterium, Lactobacillus*, *Ruminococcaceae*	Small number of examinees	[13] Wang et al., 2020.
39 CSU patients compared to 40 HCs	Fecal samples were analysed using 16S rRNA gene sequencing and untargeted metabolomics.	A significant difference in beta diversity with no significant differences in alpha diversity between CSU patients and HCs (*p* < 0.05). Gut metabolomics showed decreased isobutyric acid in CSU patients.	Phylum *Firmicutes, Bacteroidetes, Proteobacteria, Actinobacteria* Genus *Lactobacillus*, *Turicibacter*, *Lachnobacterium*	Genus *Phascolarctobacterium*	Single-center study, small number of examinees, confounding factors (age and diet) not taken into account	[17] Wang et al., 2021.
20 CSU patients compared to 20 age- and sex-matched HCs	Fecal microbial composition was analysed using 16S rRNA sequencing.	Beta diversity significantly differed between the two groups while alpha diversity did not.	Phylum *Firmicutes, Bacteroidetes, Proteobacteria* (*p* = 0.03)*, Verrucomicrobia* Class *Bacilli* (*p* = 0.04) Order *Enterobacterales* (*p* = 0.03) Family *Enterobacteriaceae* (*p* = 0.03)	Genus *Megamonas, Megasphaera*, *Dialister* (all *p* < 0.05)	Small number of examinees	[44] Zhang et al., 2021.
25 CSD patients compared to 25 age- and sex-matched HCs	Fecal samples were analysed by 16S rRNA sequencing, additionally verified by qPCR.	There were significant differences in both alpha and beta diversity in different indices with *Subdoligranulum* and *Ruminococcus bromii* (short chain fatty acid producing bacteria) as the gut microbiota biomarkers in CSD.	Phylum *Fusobacteria* Class *Gammaproteobacteria, Fusobacteria* *Order Enterobacteria, Fusobacteria* Family *Enterobacteriaceae, Fusobacteraceae, Peptostreptococcaceae, Streptococcaceae* Genus *Klebsiella*	Phylum *Firmicutes* Class *Clostridia, Alphaproteobacterial, Deltaproteobacteria* Order *Clostridiales, Rhodospirillales, Caulobacterales, Desulfovibrionales* Family *Ruminococceae, Rikenellaceae, Muribaculaceae, Christensenellaceae, Caulobacteraceae* *Genus/species Subdoligranulum, Ruminococcus bromii*	Small number of examinees	[56] Liu et al., 2021.
25 CSU patients with nsAH resistance, 19 CSU patients without nsAH resistance and 19 HCs	16S rRNA sequencing of the intestinal microbiome.	Much lower alpha diversity and evenness was observed in CSU patients with nsAH resistance than in those without (*p* < 0.05), who together with HCs showed almost no change in genera bacterium.	Genus *Prevotella*, *Megamonas*, *Escherichia*, *Succinivibrio*, *Klebsiella*, *Colidextribacter* (all *p <* 0.05) (in nsAH-resistant patients)	Genus *Blautia*, *Alistipes*, *Anaerostipes*, *Lachnospira* (all *p <* 0.05) (in nsAH-resistant patients)	Single-center study, small number of examinees, nsAHs other than levocetirizine not included	[58] Song et al., 2022.
15 responders and 15 non-responders to nsAH with CSU, with an extended cohort of another 30 responders and 30 non-responders	Gut microbiota (fecal samples) were analysed using 16S rDNA sequencing (verified by qPCR).	The abundance of *Lachnospira* in responders was higher than in non-responders with possible prediction of the AH efficacy in CSU patients.		Family *Lachnospiraceae* Genus *Lachnospira*	Small number of examinees	[59] Liu et al., 2022.

Abbrevations: CU—chronic urticaria; CSU—chronic spontaneous urticaria; CSD—symptomatic dermographism; HC—healthy controls; nsAH—second-generation non-sedating H1—antihistamines; qPCR—quantitative polymerase chain reaction; rDNA—recombinant DNA; rRNA—ribosomal RNA.

## Data Availability

The data that support the findings of this study are openly available in PubMed or available in other sources.

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
