# Peer review of "Gut Microbiome Composition in Patients with Chronic Urticaria: A Review of Current Evidence and Data"

_life, 2023, doi:10.3390/life13010152_

Round 1
Reviewer 1 Report
It is an interesting article on gut microbiome and chronic urticaria. In general, it is well written and has good information, although there is some repetitive information between different sections of the paper.
Author Response
Dear Editor and Reviewers,
thank You for Your valuable time and useful contribution You have put into assessing our previous version of the manuscript entitled „GUT MICROBIOME COMPOSITION IN PATIENTS WITH CHRONIC URTICARIA: A REVIEW OF CURRENT EVIDENCE AND DATA”. We have the consideration to each comment as follows:
REVIEWER 1
-It is an interesting article on gut microbiome and chronic urticaria. In general, it is well written and has good information, although there is some repetitive information between different sections of the paper.
Thank You! We changed some parts to avoid repetitions.
Thank You again!
Sincerely,
Authors

Reviewer 2 Report
Title of the article:
Gut microbiome composition in patients with chronic spontaneous urticaria: a review of current evidence and data
„Chronic spontaneous urticaria (CSU) is one of the most common dermatologic diseases and is defined by continuous or intermittent presence of wheals for a period exceeding 6 weeks, without an identifiable trigger.
Although the etiopathogenesis is unclear, CSU is considered to be an immune-mediated inflammatory disease, which can be triggered by various factors, such as autoimmunity, autoallergy, and imbalances in inflammation and coagulation factors.
Recent studies have linked gut microrganism composition and urticaria; however, the underlying mechanisms responsible for this connection are unknown.
Since the human immune system is in homeostasis with microbiota, and the composition of the microbiome regulates development and function of the immune system, it is likely that an alteration of microbiota components could influence the course of CSU, including disease severity, patient quality of life and treatment outcome.
To date, several studies have identified changes in the gut microbiota composition of patients with CSU, though only a few exhibited metabolic abnormalities associated with gut dysbiosis.
Further research in this field could improve clinical, diagnostic, and therapeutic approaches to patients with CSU. By applying new knowledge on gut microbial communities and metabolomics, future CSU therapies could modify the microbiota composition using agents such as probiotics or other similar agents, which, in combination with current standard therapies, could hopefully lead to a reduction in symptoms and an improved quality of life for CSU patients.”
My opinion
The aticle is well - organised and written. Introduction, methods, results and discussion are written in a clear and sound manner.
There are some typing errors that should be corrected.
The article is interesting, gives the new information on the topic of gut microbiome in patients with chronic spontaneous urticaria.
Author Response
Dear Editor and Reviewers,
thank You for Your valuable time and useful contribution You have put into assessing our previous version of the manuscript entitled „GUT MICROBIOME COMPOSITION IN PATIENTS WITH CHRONIC URTICARIA: A REVIEW OF CURRENT EVIDENCE AND DATA”. We have the consideration to each comment as follows:
REVIEWER 2
-The aticle is well-organised and written. Introduction, methods, results and discussion are written in a clear and sound manner. There are some typing errors that should be corrected.
In the new version they are corrected. Also, it was corrected by a native speaker.
-The article is interesting, gives the new information on the topic of gut microbiome in patients with chronic spontaneous urticaria.
Thank you!
Sincerely,
Authors

Reviewer 3 Report
The manuscript presents the role of the gut microbiota in the pathogenesis of chronic spontaneous urticaria. This is a very up-to-date topic and I am glad the authors decided to review the current insights.
Below I present some suggestions:
1. The title implies the authors will mostly focus on chronic spontaneous urticaria, while many cited studies analyzed different forms of chronic urticaria - please consider changing the title to be more general.
2. P. 2, line 55 - the authors state that steroids, cyclosporine and biologics are a alternative therapeutic options in CSU, while these drugs are add-on medications to antihistamines. Please correct the line regarding the treatment to reflect the current therapeutic recommendations.
3. Although the study is not a systematic review, I believe that Materials and Methods section including searched databases and keywords would improve the scientific value of the article.
4. Some sections could be reorganized to help maintain a logical sequence of the article. For example, the information on the healthy gut microbiome and immunopathogenesis of CSU is fragmentized and introduced in different parts of the article. I encourage the authors to provide separate sections discussing the physiological composition of the core microbiota and immunopathogenesis of CSU to provide the reader with reference for microbiota alterations in CSU and its possible pathogenetic implications on the immunology of this disease. Also, please provide a separate section for therapeutic implications (currently, it is also fragmentized in the text).
5. Table 1 is hard to read. Please provide separate columns for the study group, methodology, main findings and limitations of the cited studies. Please explain the abbreviations in the Table legend.
6. P. 7, line 228 - I believe the authors meant the Treg function is enhanced, not impaired, please verify.
7. P. 7, line 232 - in many cases of CSU the level of IgE is within normal limits - how would the authors then explain this pathway in CSU? Please elaborate.
Author Response
Dear Editor and Reviewers,
thank You for Your valuable time and useful contribution You have put into assessing our previous version of the manuscript entitled „GUT MICROBIOME COMPOSITION IN PATIENTS WITH CHRONIC URTICARIA: A REVIEW OF CURRENT EVIDENCE AND DATA”. We have the consideration to each comment as follows:
REVIEWER 3
-The manuscript presents the role of the gut microbiota in the pathogenesis of chronic spontaneous urticaria. This is a very up-to-date topic and I am glad the authors decided to review the current insights. Below I present some suggestions:
-1. The title implies the authors will mostly focus on chronic spontaneous urticaria, while many cited studies analyzed different forms of chronic urticaria - please consider changing the title to be more general.
In the new version this is corrected and now the title includes term„“chronic urticaria“.
-2. P. 2, line 55 - the authors state that steroids, cyclosporine and biologics are alternative therapeutic options in CSU, while these drugs are add-on medications to antihistamines. Please correct the line regarding the treatment to reflect the current therapeutic recommendations.
In the new version this is also corrected. Thank You!
-3. Although the study is not a systematic review, I believe that Materials and Methods section including searched databases and keywords would improve the scientific value of the article.
In the new version this is corrected and this information was added.
-4. Some sections could be reorganized to help maintain a logical sequence of the article. For example, the information on the healthy gut microbiome and immunopathogenesis of CSU is fragmentized and introduced in different parts of the article. I encourage the authors to provide separate sections discussing the physiological composition of the core microbiota and immunopathogenesis of CSU to provide the reader with reference for microbiota alterations in CSU and its possible pathogenetic implications on the immunology of this disease. Also, please provide a separate section for therapeutic implications (currently, it is also fragmentized in the text).
In the new version this is corrected and we changed some parts of the text. Also, we added separate sections for physiological composition of the core microbiota (healthy persons), immunopathogenesis of CSU and therapeutic implications
-5. Table 1 is hard to read. Please provide separate columns for the study group, methodology, main findings and limitations of the cited studies. Please explain the abbreviations in the Table legend.
In the new version of table, this is changed/corrected and the new table includes more columns with specific information, as mentioned by the reviewer.
-6. P. 7, line 228 - I believe the authors meant the Treg function is enhanced, not impaired, please verify.
In the new version this is corrected. Thank You!
-7. P. 7, line 232 - in many cases of CSU the level of IgE is within normal limits - how would the authors then explain this pathway in CSU? Please elaborate.
In the new version the explanation for this dana is added to the text and supported by reference (Schmetzer,2018), page 8.
Thank You again!
Sincerely,
Authors